# Design of a Low Scattering Metasurface for Stealth Applications

**DOI:** 10.3390/ma12183031

**Published:** 2019-09-18

**Authors:** Tayyab Ali Khan, Jianxing Li, Juan Chen, Muhammad Usman Raza, Anxue Zhang

**Affiliations:** 1School of Electronic and Information Engineering, Xi’an Jiaotong University, Xi’an 710049, China; khantayyabali@xjtu.edu.cn (T.A.K.); anxuezhang@xjtu.edu.cn (A.Z.); 2Department of Electrical Engineering, City University of Hong Kong, Hong Kong SAR 999077, China; 3Shenzhen Research School, Xi'an Jiaotong University, Shenzhen, Guangdong 518057, China; 4Guangdong Xi'an Jiaotong University Academy, Fushan, Guangdong 528300, China

**Keywords:** metasurface, metamaterial, radar cross section (RCS)

## Abstract

The design of a metasurface with low radar cross section (RCS) property is presented in this paper. The low scattering of the metasurface is achieved by applying the artificial magnetic conductor (AMC) unit cells in three different configurations. Two different AMC unit cells with an effective phase difference of 180 ± 37° are first designed to analyze the out of phase reflection in a wideband frequency range from 5.9 to 12.2 GHz. Then, the unit cells are placed in a chessboard-like configuration, newly constructed rotated rectangular-shaped configuration, and optimized configuration to study and compare the RCS reduction performance. All designs of the metasurface with different configurations show obvious RCS reduction as compared with the metallic plate of the same size. However, the relative bandwidth of the optimized metasurface is larger than the chessboard-like configuration and rotated rectangular-shaped configuration. To certify the results of the simulations, the metasurface with the optimized configuration is fabricated further to measure the RCS reduction bandwidth. The measured results are in good accordance with the simulated results. Therefore, the proposed metasurface can be a good option for applications where low RCS is desirable.

## 1. Introduction

With the rapid advancements in detection and stealth technology, the stealth performance of military targets is highly desirable. The scattering from the surface of the targets is generally defined in terms of the radar cross section (RCS). Broadband RCS reduction is necessary for improving the stealth performance of the system. In the recent past, researchers have shown great interest in reducing the RCS of surfaces to achieve low observability from radars. There are four main techniques extensively used in literature for the reduction of the target RCS, including radar absorbing materials (RAMs), reshaping of the target surface, use of active cancellation, and passive cancellation methods [1].

RAMs use an absorption mechanism to reduce the reflected energy from the target surface. The absorption mechanism is due to the several losses produced by the dielectric or magnetic properties of the material [2]. In [3,4], the RCS reduction of the different surfaces has been obtained using RAMs, but it was concluded that the absorbing behavior of the surface is in a narrow bandwidth. By using the reshaping technique, the surface of the target can be modified to redirect the incoming electromagnetic energy (EM) away from the source. The primary drawback of reshaping the surface is that it may alter the other performance parameters of the surface [5]. Furthermore, all these practical techniques reduce the RCS of the targets in a narrow bandwidth. In [6], the arrangement of the artificial magnetic conductor (AMC) and perfect electric conductor (PEC) is employed for destructive interference in the normal direction. The interference of PEC and AMC reflection signals helps to disperse the backscattered energy to reduce the RCS. However, due to the out of phase reflection behavior of the PEC and AMC, RCS reduction in a narrowband has been realized. To overcome this issue, the combinations of AMCs with different configurations [7,8] or different sizes [9,10] have been proposed to fulfill the phase cancellation criteria. The idea is to utilize the out of phase reflection of the unit cells to achieve effective cancellation over a wide frequency band. Furthermore, these unit cells have been arranged in a chessboard-like configuration to scatter the reflected energy into four diagonal directions with low scattered energy in the normal direction [11,12,13]. In [14,15], the unit cells of different sizes have been arranged randomly to reduce the scattered energy from the surface to the other directions. Recently, a new concept called coding metamaterials for the manipulation of EM waves from surfaces has received the attention of researchers [16,17,18]. To obtain the optimized coding surface with minimum scattered energy, many optimization methods have been applied. However, these optimization methods need time to generate the optimal coding sequences for the optimized coding surface [19]. Therefore, the design of surfaces with high RCS reduction bandwidth and low complexity is highly desirable.

In this work, three different metasurfaces are implemented to study the effect of different patterns of the AMC unit cells on the RCS reduction bandwidth. The AMC unit cells are arranged in a chessboard-like configuration, rotated rectangular-shaped configuration, and optimized configuration. The analysis shows that the optimized configuration of metasurface produces the widest RCS reduction bandwidth of over 95% in relative bandwidth. The complete design and analysis of the chessboard-like configuration, rotated rectangular-shaped configuration, and optimized configuration are presented in this paper. In the end, the optimized configuration of the metasurface is manufactured and tested to validate the simulation results. The measured results show a good correspondence with the simulated results.

## 2. Design of the AMC Unit Cells

The low scattering performance of the surfaces under normal incidence can be achieved by redirecting the incident EM energy to the other directions. The reflections from the surface depend on the phase profile of the surface. The PEC surface shows the strongest reflection upon the incidence of the plane waves. These strongest reflections limit the use of PEC for stealth applications where low observability is vital. Therefore, it is important to study the design of a surface which could minimize the scattered energy from the surface. The RCS reduction from the surface compared to the PEC can be expressed as Equation (1) [15]:(1)$$\sigma (dBsm)=10\log\left[\frac{\underset{r\to \infty }{\lim}4\pi r^{2}{\left|\frac{E_{s}}{E_{i}}\right|}^{2}}{\underset{r\to \infty }{\lim}4\pi r^{2}{\left(1\right)}^{2}}\right]=10\log\left[{\left|\frac{E_{s}}{E_{i}}\right|}^{2}\right]$$

According to Equation (1), *Es* and *Ei* represent the scattering and incident electric fields, respectively, whereas *σ* and *r* depict the radar cross section reduction and far-field distance respectively. In order to reflect the incident wave to the other directions, the reflection phase of the surface needs to be optimized in such a way that it should not retransmit the EM energy into the specular direction. For the case of a metallic surface, the constant phase throughout the surface strongly reflects the incident energy back to the incident directions. The phase profile of the surface needs to be controlled to produce the out of phase reflections. Therefore, two different AMC unit cells with wideband out of phase reflection property are designed. The top and side views of the unit cells are presented in Figure 1. The structure of the unit cells consists of the square metallic patches of different sizes. The out of phase reflections from the surface would allow redirecting the scattering lobes away from the normal direction. The Rogers R3003 (Rogers corporation, Xi'an, China) with a relative permittivity of 3.0 and thickness h = 3.18 mm was selected as a substrate. Both the unit cells are backed by a metallic ground to minimize the transmission. Due to less transmission and strong reflections, the absorption of the EM energy inside the surface would be much less.

The frequency domain solver of the computer simulation technology (CST) software version 17 (Dassault Systèmes, Vélizy-Villacoublay, France) was used to design and optimize the geometric parameters of the unit cells. The following optimal parameters were obtained: W = L = 6 mm, W1 = L1 = 5.6 mm, W2 = L2 = 2.7 mm. The total reflection from the surface having two different AMCs, upon the incidence of a plane wave from normal direction, can be given as an average reflection from both the unit cells [15]. The RCS reduction in dBsm can be given as Equation (2):(2)$$\sigma (dBsm)=10\log\left[{\left|\frac{B_{1}e^{j\theta_{1}}+B_{2}e^{j\theta_{2}}}{2}\right|}^{2}\right]$$

According to Equation (2), *θ*_1_ and *θ*_2_ represent the reflection phase of the unit cell 1 and unit cell 2, respectively. Furthermore, *B*_1_ and *B*_2_ represent the reflection magnitude of the unit cell 1 and unit cell 2, respectively. When the reflection phase difference (*θ*_2_ − *θ*_1_) between the unit cells becomes ±180°, the scattered energy from the surface becomes minimum. According to [15], the 10 dB RCS reduction can be obtained, if the phase difference between the unit cells follows 180 ± 37° condition. Figure 2 shows the reflection phase performance of the designed unit cells. It can be seen that the two different unit cells exhibit zero reflection phase at different frequencies and an effective phase difference following the 180 ± 37° condition from 5.9 to 12. 2 GHz. Therefore, a 10 dB RCS reduction can be expected in this range. A low RCS of the surface can be anticipated due to the redirection of the incident energy to other directions. 

## 3. Design of Differently Configured Metasurfaces 

To meet the periodic boundary condition (PBC), the periodic array consists of 5 × 5 elements of unit cell 1 and unit cell 2 are designed and named as ‘0’ and ‘1’ element, respectively. The length of ‘0’ and ‘1’ is generally selected as about one wavelength. Thus, in this work, we took 5 × 5 unit cells as an element. The principle of obtaining the full surface is associated with the placement of element ‘1’ and ‘0’. Here, we consider three different cases for the placements of ‘1’ and ‘0’. Firstly, the elements are placed in a chessboard-like configuration to get the low observable surface. Secondly, the AMC unit cells are arranged in the rotated rectangular-shaped structure to analyze the RCS reduction bandwidth. Thirdly, we optimize the placement of ‘1’ and ‘0’ elements to produce the best possible solution for low RCS of the surface. 

The design of a metasurface based on a chessboard-like configuration is depicted in Figure 3. The dimension of the surface is 180 mm × 180 mm. The arrays of ‘0’ and ‘1’, which are composed of 5 × 5 AMC blocks of unit cell 1 and unit cell 2, respectively, were arranged in a chessboard-like configuration to study the RCS performance. The direction of scattering lobes from the surface depends on the reflection phase profile of the surface. Therefore, it is very important to study the effect of the arrangement of ‘0’ and ‘1’ array on the scattering performance of the surface.

To demonstrate the scattering analysis of the chessboard-like surface, a metallic plate of the same size is applied as a reference to compare the RCS performance of the chessboard-like surface. Figure 4 shows the scattering behavior of the metallic plate and the chessboard-like surface. The scattering analysis of the metallic plate and chessboard-like surface are carried out using the time domain solver of the CST software. Figure 4a depicts the strong reflections from the surface of the metallic plate when a plane wave impinges from the normal direction. The metallic plate has a constant reflection phase over the surface. Therefore, the resonant waves from the surface show a strong reflection mainly in a single lobe. In the case of the chessboard-like surface, due to the different reflection phases of arrays, the resonant waves from the surface are scattered into four directions, as can be seen in Figure 4b. This further shows that a lower amount of energy is reflected back towards the normal direction as all the reflected waves are scattered into four directions. Figure 4c shows the monostatic RCS performance of the reference metallic plate and proposed chessboard-like surface. Obvious RCS reduction can be seen in a wideband as compared to the reference surface. The maximum RCS reduction of 22 dB can be seen at 11.9 GHz, where the two AMCs show a maximum out of phase reflection. Figure 4d depicts the RCS reduction curve. The RCS reduction from 5.5 to 14 GHz is obvious and validates the low scattering property of the chessboard-like surface. However, the 10 dB RCS reduction is separated into three bands from 6.2 to 6.9 GHz, 7.8 to 8.2 GHz, and 10.6 to 12.3 GHz. In order to improve the 10 dB RCS reduction bandwidth, it is required to investigate the optimal configuration of AMCs that could produce less scattering amplitude towards the normal direction and increase the relative bandwidth of the RCS reduction. 

In the second model, a rotated rectangular-shaped configuration was adopted to study the effect of different placement of AMCs on the RCS reduction bandwidth. The metasurface based on the rotated rectangular-shaped configuration of AMCs is presented in Figure 5. Four arrays consisting of different AMC patterns were utilized to construct the surface of the rotated rectangular-shaped metasurface. The size of the newly constructed rotated rectangular-shaped metasurface is the same as for the chessboard-like surface. Due to the reflection between the two different types of AMCs within a single array element, it is expected that the overall layout of the metasurface would generate a more diffused reflection pattern as compared to the chessboard-like surface. Hence, the more diffuse patterns would lead to less scattering in the normal direction and increase the RCS reduction bandwidth.

To observe the scattering behavior of the newly constructed rotated rectangular-shaped metasurface, RCS analysis using the time domain solver of the CST software was performed. The scattering performance of the rotated rectangular-shaped metasurface compared with the same size of the metallic plate is given in Figure 6. It can be seen from Figure 6a,b that as compared to the metallic plate which strongly reflects the incident energy back to the incident direction in a single lobe, the newly constructed rotated rectangular-shaped metasurface redirects the incident plane waves to the other directions, i.e., mainly in eight lobes. Due to the two different phase profiles offered by each array of the metasurface, the overall metasurface redirects the incident energy into more directions as compared to the chessboard-like configuration. Therefore, a lower amount of scattering energy as compared to the chessboard-like configuration was redirected to the normal direction by the surface. Figure 6c depicts the monostatic RCS performance comparison of the metallic plate and newly constructed rotated rectangular-shaped metasurface. Compared to the metallic plate of the same size, a significant RCS reduction is evident for the newly constructed rotated rectangular-shaped metasurface. The maximum RCS reduction of 22.5 dB at 6.3 GHz can be clearly seen from Figure 6c. The RCS reduction curve is discussed in Figure 6d. The noticeable RCS reduction for the rotated rectangular-shaped metasurface is clearly shown from 5 to 14 GHz. However, the 10 dB RCS reduction is from 5.8 to 7.5 GHz and 8.75 to 12.4 GHz. From Figure 6d, it can be concluded that the 10 dB RCS reduction bandwidth has increased for the case of the rotated rectangular-shaped metasurface as compared to the chessboard-like surface.

The above analysis shows that the placement of AMC unit cells—to construct the low scattering surface—affect the RCS reduction bandwidth. The chessboard-like surface and newly constructed rotated rectangular-shaped surface scatter the reflected energy into four and eight lobes, respectively. Hence, more scattering lobes from the surface would allow the reflected energy to redirect into more directions and the scattering amplitude towards the normal direction would reduce. Therefore, to enhance the diffusion among the scattered lobes, the arrays of ‘0’ and ‘1’ were arranged randomly, and the RCS performance calculated. To find the best possible solution of a random matrix for low RCS, an optimization algorithm was applied, and far-field radiation patterns from the surface were calculated using MATLAB. The optimized layout with diffused radiation patterns and low scattering amplitude towards the normal direction were considered for the study of RCS reduction bandwidth.

The arrays of ‘0’ and ‘1’ produce a progressive phase shift within 180 ± 37° and can be considered as planar propagating arrays. The phase patterns of each array inside the surface can be optimized by array factor theory [19]. The total scattering field of the metasurface can be represented by the Y × Z matrix of ‘0’ and ‘1’, which can be further optimized to get the optimal matrix sequences. The array factor of the surface consists of ‘0’ and ‘1’ array elements can be given as Equation (3):(3)$$AF=\sum_{y=1}^{Y}\sum_{z=1}^{Z}\exp\{j\{\left(y-\frac{1}{2}\right)\left(kd\sin\theta \cos\phi \right)+\left(z-\frac{1}{2}\right)\left(kd\sin\theta \sin\phi \right)+\phi \left(y,z\right)\}\}$$

According to Equation (3), theta (*θ*) and phi (Φ) depict the elevation and azimuth angles, respectively; whereas *d* and φ(y,z) show the distance between the arrays of ‘0’ and ‘1’ and the initial phase of the lattice, respectively. For low RCS performance of the metasurface, good diffusion performance is required. In order to diffuse the scattering wave as much as possible, the peak value of the scattering field is employed as the fitness function. Therefore, the fitness function can be given as Equation (4):(4)Fitness=max[AF(θ,ϕ)]

It is important to note that the max [AF(θ,Φ)] expression indicates the maximum value of scattered energy in the normal direction. For optimization, our goal is to find the best matrix that could produce the minimum value of scattered energy. Therefore, a genetic algorithm using MATLAB was applied to find out the best optimal matrix with minimum scattered energy. The flowchart of the algorithm is presented in Figure 7. The initial parameters, such as the size of the metasurface, working frequency, wave number, etc., are set at the start to initiate the algorithm. In the first step, a random initial population of ‘0’ and ‘1’ sequences are generated and the fitness value corresponding to the initial population is calculated. In the second step, the new random sequence is generated, and its fitness value is compared with the previously available value. The coding matrix with minimum fitness function value is considered for the optimized metasurface. To find the best possible matrix with minimum fitness value, the algorithm is repeated 1000 times. After obtaining the best possible matrix from the optimization, the arrays of ‘0’ and ‘1’ are arranged according to the matrix to obtain the layout of the optimized metasurface. The structure of the optimized metasurface is shown in Figure 8. 

To validate the RCS performance of the optimized metasurface, it was compared with the metallic plate as shown in Figure 9. Due to the constant reflection phase of the metallic plate, strong reflection can be seen in Figure 9a. However, for the case of the optimized metasurface, all the reflected energy from the surface was diffused in different directions due to multiple reflections among the arrays of the surface. The diffusion of reflected resonant waves is depicted in Figure 9b. The scattering amplitude of the reflected waves was rather low in the normal directions. The monostatic RCS performance, compared with the metallic plate, is shown in Figure 9c. Compared to the metallic plate, the optimized metasurface has a low scattering property from 5 to 14 GHz. The 10 dB RCS reduction is obvious from 5.9 to 12.2 GHz, which is much wider as compared with the chessboard-like and rotated rectangular-shaped surface. The RCS reduction performance of the optimized metasurface is further explained in Figure 9d. 

The bistatic RCS performance at 8 GHz of the chessboard-like surface, rotated rectangular-shaped surface, and optimized metasurface is presented in Figure 10. In XZ and YZ planes, the RCS reduction can be seen in the angular region of −20° ≤ *θ* ≤ −20° for all three configurations. A good bistatic RCS performance of the optimized metasurface is limited in the small angular region −10° ≤ *θ* ≤ −10°, as compared to the chessboard surface. However, the optimized metasurface has a less scattering amplitude as compared to the chessboard-like surface and rotated rectangular-shaped surface in the normal direction. 

## 4. Fabrication and Measurements

To verify the scattering performance of the designed metasurfaces through experiment, only the proposed metasurface with optimized configuration of designed AMCs was fabricated and measured. The PEC surface with the same size as the designed optimized metasurface was considered as a reference surface to compare the scattering performance. The photograph of the fabricated optimized metasurface is depicted in Figure 11. Limited by the experimental amenities, the RCS was measured in terms of reflectivity to validate the simulated results. The measurement arrangement to obtain reflections from the surface of the reference and proposed surface in the monostatic configuration is shown in Figure 11. The reflectivity of reference and the proposed surface was measured with the help of two horn antennas used as a transmitter and a receiver. The transmitter is placed in the normal direction to the surface of the reference and proposed surfaces. The reflections from the surface of the reference and proposed surface are captured by the receiver in a monostatic configuration. 

The comparison between the simulated and measured RCS reduction is demonstrated in Figure 12. It is evident that simulated and measured results have good accordance with each other. An obvious RCS reduction from 5 to 13 GHz is clearly shown. However, the 10 dB RCS reduction was from 5.9 to 12.2 GHz. The discrepancies between the simulated and measured results are more likely due to the fabrication and measurements tolerances. Fabrication tolerances mainly depend upon the substrate’s dielectric constant and loss tangent. Compared to the simulated case, the physical substrate may have some approximate value of dielectric constant and loss tangent. The measurement error deals with the tolerances of the measurement apparatus. For example, to measure the RCS of the fabricated prototype at normal incidence, the designed surface should present at the normal incidence of the transmitter and receiver. Due to the limited testing facilities, two horn antennas were placed at normal incidence to the designed surface to transmit and capture the reflections from the surface. The location of the receiver is very important for capturing the reflection from the surface. The small error in placement of the receiver could lead to approximate results of RCS. From the above analysis, it can be concluded that the proposed optimized metasurface has a low scattering property in wide bandwidth, and it can potentially replace the metallic surface for applications where there is a requirement for high stealthiness.

## 5. Conclusions

This paper discusses the design of a low scattering metasurface, which is highly desirable for stealth applications. Three different designs of metasurfaces, including chessboard-like surface, rotated rectangular-shaped surface, and optimized surface, were presented using two different combinations of AMC unit cells. The effect of different placements of AMCs on the RCS reduction bandwidth was explored. The analysis shows that the optimized metasurface diffused the scattering energy and had a much wider RCS reduction bandwidth as compared to the chessboard-like surface and rotated rectangular-shaped surface. The optimized metasurface was further fabricated and measured to validate the low observable property. The simulated results have good correspondence with the measured results. Therefore, the optimized metasurface can be a good candidate for applications where low RCS in wideband is required.

## Figures and Tables

**Figure 1 materials-12-03031-f001:**
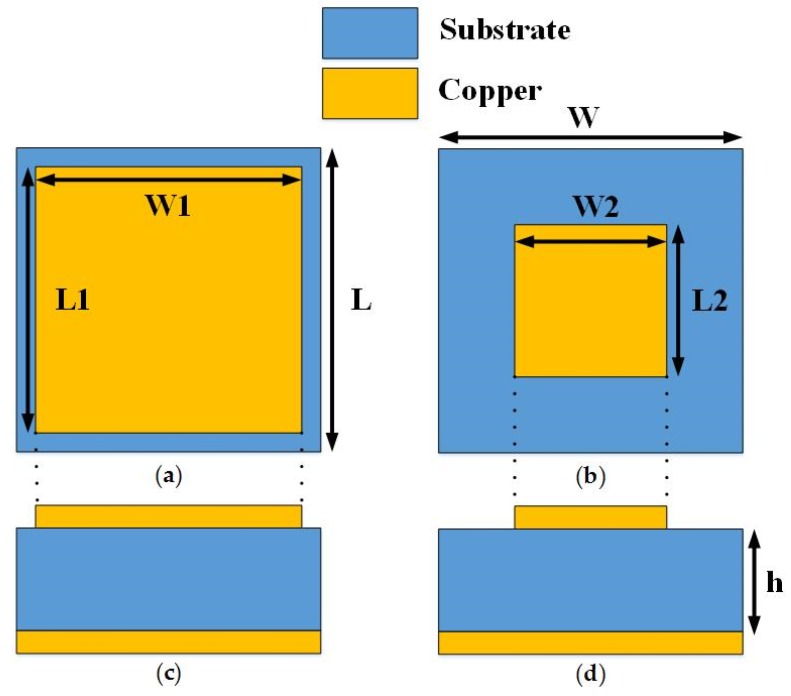
The artificial magnetic conductor (AMC) unit cells. (**a**) Top view of the unit cell 1; (**b**) Top view of unit cell 2; (**c**) Side view of the unit cell 1; (**d**) Side view of the unit cell 2.

**Figure 2 materials-12-03031-f002:**
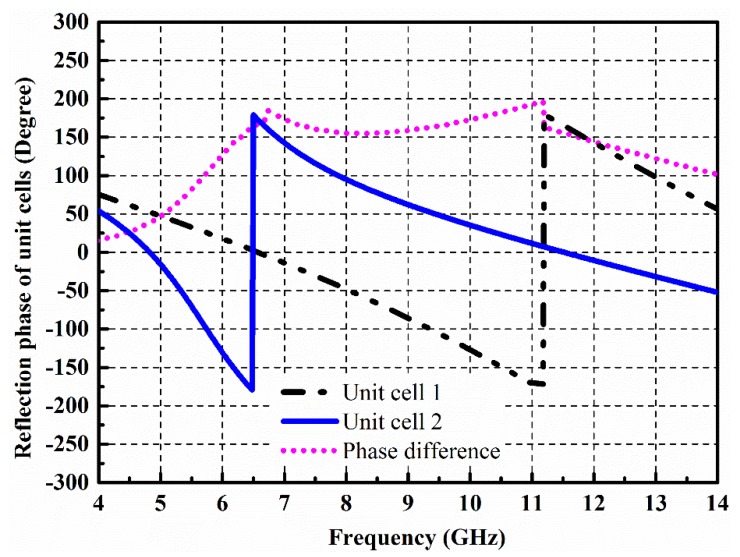
Reflection phase performance of the AMC unit cells.

**Figure 3 materials-12-03031-f003:**
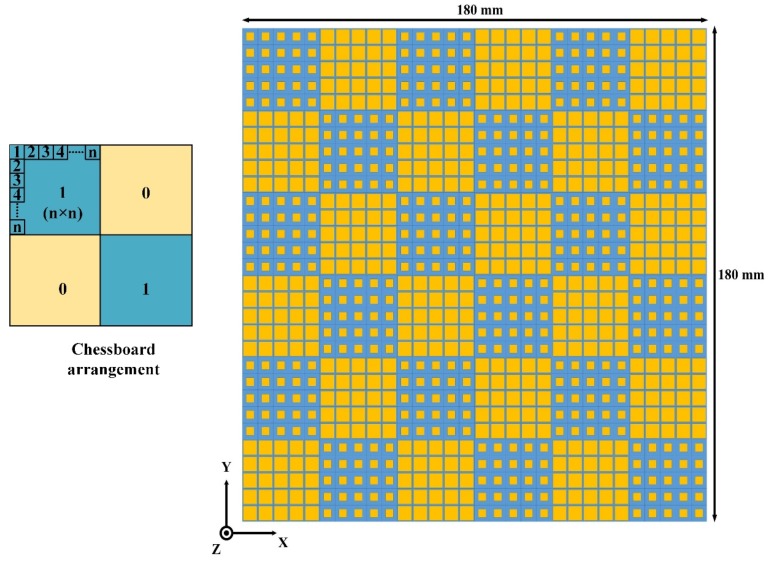
Design of the chessboard-like metasurface.

**Figure 4 materials-12-03031-f004:**
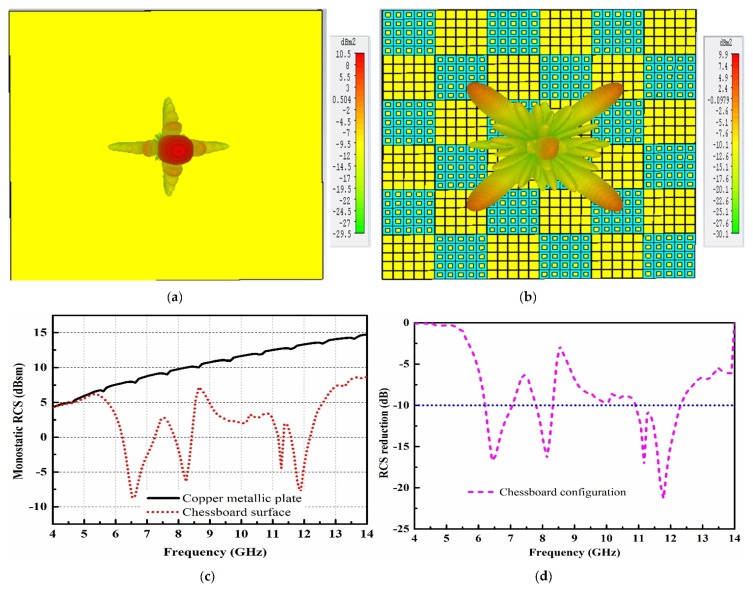
Scattering performance of the copper metallic plate and chessboard-like surface. (**a**) Scattering lobes from the copper metallic plate; (**b**) Scattering lobes from the chessboard-like surface; (**c**) Monostatic radar cross section (RCS) performance; (**d**) The RCS reduction performance.

**Figure 5 materials-12-03031-f005:**
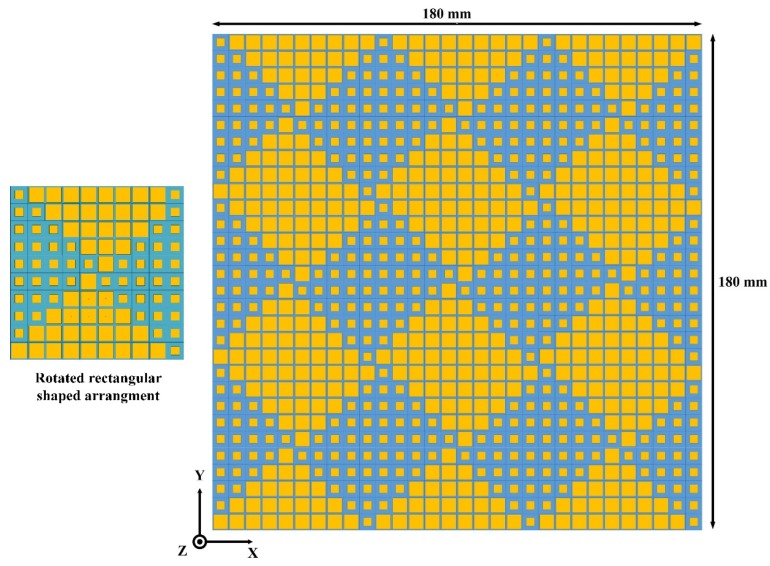
Design of the rotated rectangular-shaped metasurface.

**Figure 6 materials-12-03031-f006:**
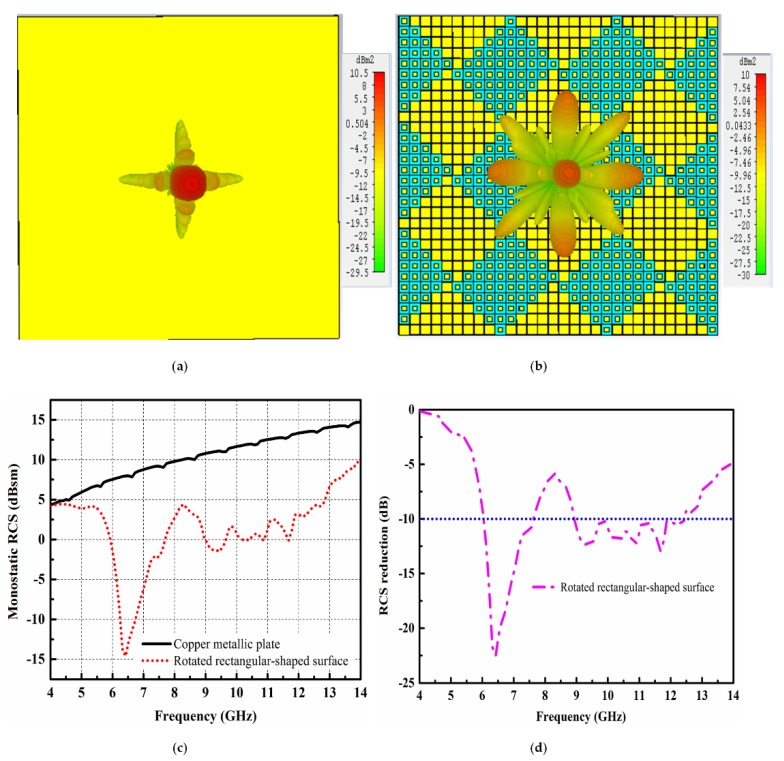
Scattering performance of the copper metallic plate and rotated rectangular-shaped surface. (**a**) Scattering lobes from the copper metallic plate; (**b**) Scattering lobes from the rotated rectangular-shaped surface; (**c**) Monostatic RCS performance; (**d**) RCS reduction performance.

**Figure 7 materials-12-03031-f007:**
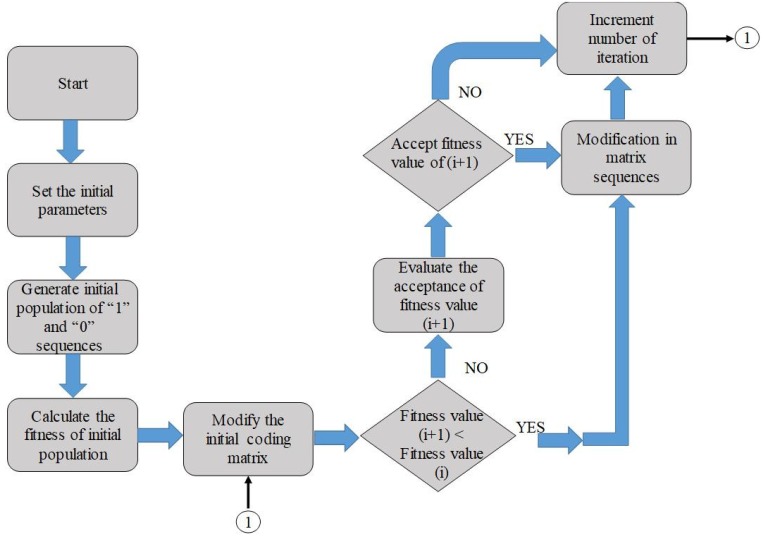
The optimized algorithm.

**Figure 8 materials-12-03031-f008:**
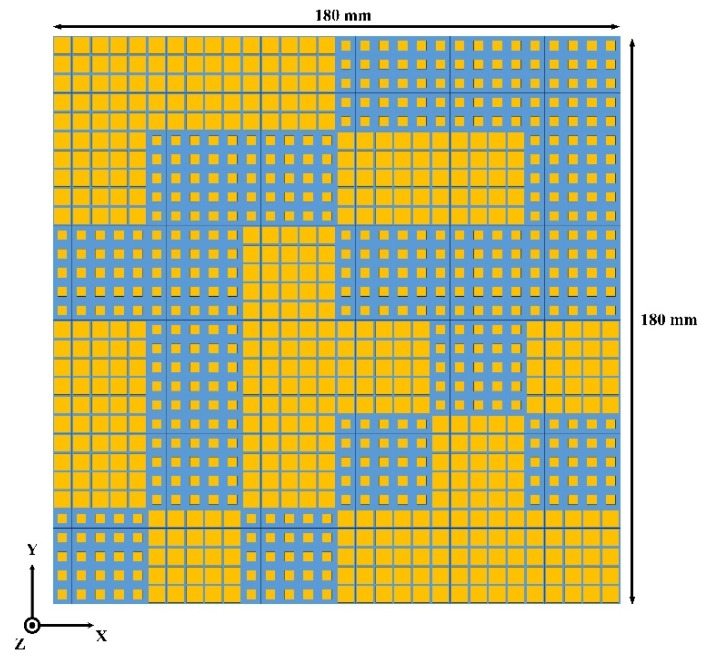
The optimized metasurface.

**Figure 9 materials-12-03031-f009:**
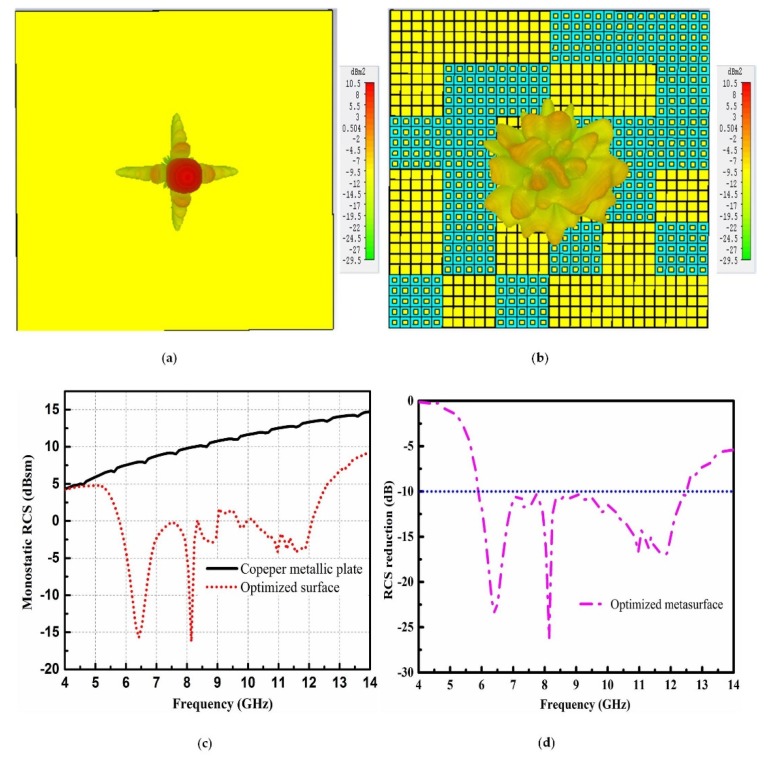
Scattering performance of the copper metallic plate and optimized surface. (**a**) Scattering lobes from the copper metallic plate; (**b**) Scattering lobes from the optimized surface; (**c**) Monostatic RCS performance; (**d**) RCS reduction performance.

**Figure 10 materials-12-03031-f010:**
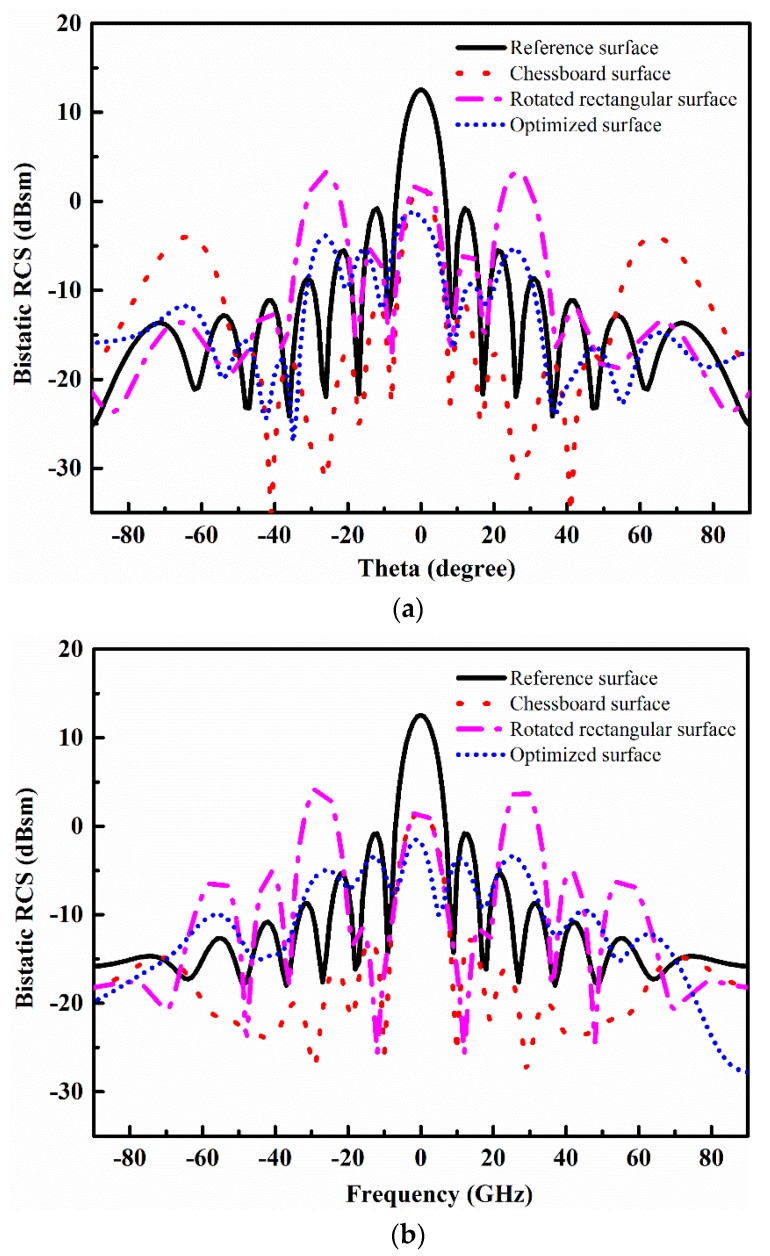
Bistatic RCS performance. (**a**) XZ plane; (**b**) YZ plane.

**Figure 11 materials-12-03031-f011:**
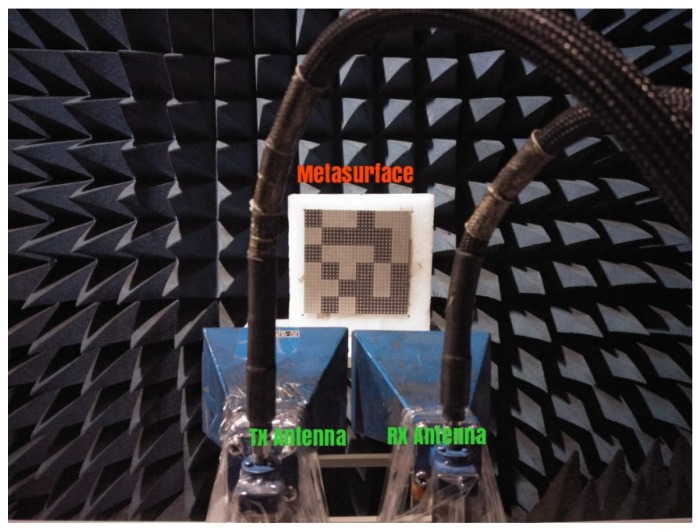
Measurement setup for the reflectivity of an optimized metasurface.

**Figure 12 materials-12-03031-f012:**
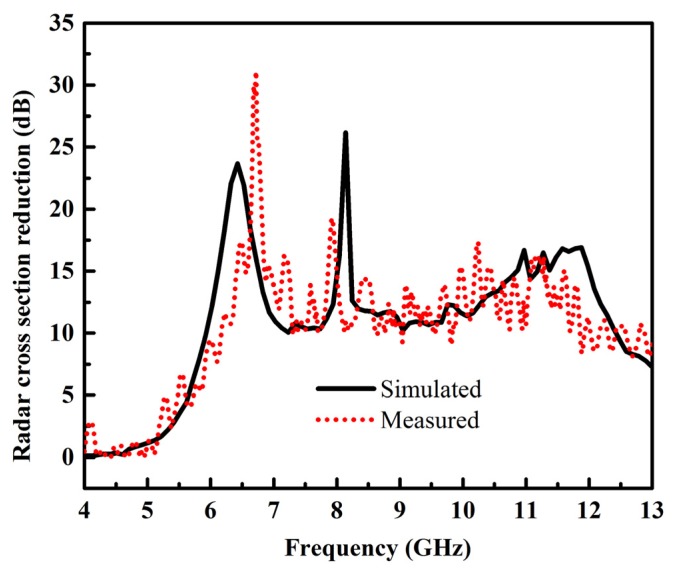
RCS reduction comparison between simulated and measured results.

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
