# Peer review of "Design of a Low Scattering Metasurface for Stealth Applications"

_materials, 2019, doi:10.3390/ma12183031_

Round 1

Reviewer 1 Report

The work has interesting points; I believe that there is significant room for this work. I do not have any problem in supporting the publication of this work, except to bring up the following a minor issue for the authors to consider before having their work published:

Authors should write the name of the metallic plate used in Figure 4 in orde to give the reader a bit clarity, also state in the reference of the Figure legend .

Author Response

Special thanks to you for your good comments.

Reviewer 2 Report

The paper is well written, well cited, and proper illustrations are used where needed, and well explained. However, it seems like the title and conclusions are true only if taken very literally, and some words are used too loosely, specifically "optimized" and "triangular".

Some small grammatical errors exist. For example, in line 81: "According to (1), Es and Ei represent...".  In equation (1), however, these are actually Es and Ei. Stricly speaking, (1) has no variables Es and Ei. Figure 5 illustrates the geometry of the triangular-shaped arrangement and metasurface -- these are not the same. The Triangular shaped arrangement shows a symmetric pattern of 4 triangles, all of which meet in the center. The metasurface implementation does not show this -- it shows rotated rectangles. Rotated rectangles would not be produced if the pattern in the Triangular shaped arrangement were tessellated -- it would show squares. This should be explained. There is an obvious difference between the RCS produced by checkerboard pattern and the triangular arrangement. So this opens up a question about the parameter space. Physically, what changes occur when you rotate the squares in a checkerboard to resemble that of the triangular arrangement? Has this been studied? Figure 6(c) shows Monostatic RCS plotted for the (flat) reference surface and the triangular-shaped surface. The reference surface shows periodic dips in the RCS, that are not explained by the authors. This should be explained. The paper presents RCS from a reference design, and 3 additional designs: chessboard (squares), "triangular" arranement, and an "optimized" design. The optimized design is obtained by use of an optimization routine to produce a random arrangement -- however, it is not explained why this is physically better. How does this particular geometry interact with the field to produce the lower RCS? This understanding could lead to some wonderful breakthroughs in understanding in this field, yet it is going unexamined. Based on point 5, I reiterate that the title, "Design of a [...] metasurface" is true only in the strictest sense -- the author is presenting a design, but through computer optimization -- no new understanding of how the candidate design is being presented. In lines 248-250, it is stated that the optimized metasurface has less scattering amplitude compared to the other designs in the normal direction. Examine Figure 10a. This is true -- but the moment you walk off normal incidence (ex. 25 deg.), the "optimized" design is not the best design. At this angle, the chessboard surface demonstrates ~-30 dBsm, while the "optimized" design demonstrates ~-5 dBsm. Pratically speaking, the source will not always strike the surface at normal incidence -- this is a special case that is easily testable in a lab. At 25 deg., "optimized" does not seem so optimal. It seems that this should be addressed -- a discussion of real-world applications would be appreciated. In section 4, experiment results are compared with simulation. It is stated that "discrepancies between the simulated and measured results are more likely due to the fabrication and measurement tolerance".  The fabricated structure is macroscopic -- why is this not further investigated? What is the largest probable source of fabricated error? What is the largest probable source of measurement error? Understanding discrepancies between experiment and simulation are very important. Look at Figure 12 -- Simulation has a large, broad peak around 6.25GHz. This maybe kinda looks like the tall peak in "measured" at 6.75GHz, except for a few main differences: (1) the experiment peak is much, much more narrow, (2) the RCS reduction is ~7 dB taller in the measured data, and (3) there's a noticeable spectral shift. This goes back to point 9 -- discrepancies need to be understood, or explained. 

Author Response

(The authors gave the same response as above.)

Reviewer 3 Report

This manuscript presents a comprehensive investigation of a low scattering metasurface design for low radar cross-section (RCS) for stealth applications. This low scattering metasurface design is achieved using an artificial magnetic conductor (AMC) unit cells in different configurations. The manuscript reports a significant reduction in RCS using AMC designed metasurfaces compared to the metallic sheets of similar sizes.

The work presented in this manuscript is interesting and a significant contribution to the field and the community. This manuscript is well written and conveys the findings in a convincing manner, and the work presented in this manuscript is relevant to the Materials journal. I have no hesitation in recommending this manuscript to be accepted for publication.

Author Response

(The authors gave the same response as above.)

Round 2

Reviewer 2 Report

The authors showed a great deal of attention to previous criticisms of their paper. The resulting paper is solid, and adds to the body of knowledge of the field. I have no issues with this paper being published.